# Spillover effects of violent attacks and COVID-19 exposure on mental health of health professionals: A two-phase quasi-natural experiments study in Northwest China

## Research Article

health professionals; mental health; violent attacks; COVID-19; organizational support

**Corresponding authors:**
Jing Guo and Benzhong Zhang;
Emails: jing624218@163.com;
zhangbzh@lzu.edu.cn

Ning Liu and Hong Qian are co-first author.

Ning Liu[1], Hong Qian[2], Ben Zhong Zhang[3] and Jing Guo[4] 

[1]School of Management, Lanzhou University, Lanzhou, 730000, P. R. China; [2]The First Hospital of Lanzhou University, Lanzhou, 730000, P. R. China; [3]School of Public Health, Lanzhou University, Lanzhou, 730000, P. R. China and [4]Department of Health Policy and Management, School of Public Health, Peking University, Beijing, 100191, P. R. China

## Abstract

The aims of this study were to examine the spillover effects of violent attacks, coronavirus disease-2019 (COVID-19) exposure, and their interactions on health professionals' mental health, and the role of organizational support in their relationships in China. A two-phase survey data ($n$ = 10,901) before and after the first outbreak of COVID-19 was integrated with regional macro data on the number of lawsuit cases of violent attacks and COVID-19 cases. Three studies were designed to isolate the general spillover impact of violent attacks on the mental health of health professionals, how COVID-19 affects the mental health of health professionals, and whether organizational support moderates the relationship between violent attacks and mental health through econometric regressions. Violent attacks and COVID-19 are negatively associated with the mental health of health professionals, and the outbreak of COVID-19 adversely deteriorates the spillover effects of violent attacks. Physicians, not nurses, are the most affected group. Better perceived support from hospitals can significantly mitigate the adverse effects of COVID-19, violent attacks, and their interactions on the mental health of health professionals. COVID-19 deteriorates the adverse effects of violent attacks on the mental health of health professionals, while better organizational support is helpful to mitigate these effects.

## Impact statements

To the best of our knowledge, this study is the first use of micro–macro data systematically estimating the spillover effects of violent attacks, coronavirus disease-2019 (COVID-19) cases, and their interaction, on the mental health of health professionals in China. It also documents the importance of hospital support in mitigating mental health deterioration in workplaces. Using data from one of the least developed provinces in China, we provide empirical evidence of adverse spillover effects of violent attacks, COVID-19, and their interaction on the mental health of health professionals in China, where support from hospitals is crucial. Most importantly, we estimate a bottom-line situation in China, and then a worse national situation can be speculated. Our findings support studies discussing the adverse effects of violent attacks or COVID-19 on the mental health of health professionals. Particularly, we empirically evidence that COVID-19 can deteriorate the adverse effects of violent attacks, where the adverse effects can spill but are not limited to those who bear the damage. In mitigating and recovering from the adverse spillover effects on the mental health of health professionals incurred by violent attacks or COVID-19 and their interaction, support from hospitals is another crucial channel other than individual-level interventions. Our experience from one of the least developed provinces of China may be nationally or even globally generalized.

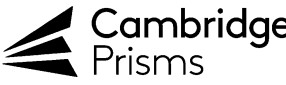



## Introduction

The increasing violent attacks against health professionals (HPs) at workplaces (or medical violence, hereinafter MV) like hospitals have gained global concern (Greenberg et al., 2020; McKay et al., 2020). Since the outbreak of coronavirus disease-2019 (COVID-19), the risk of violence in hospitals on HPs has seriously increased (Larkin, 2021; Thornton, 2022). Survey data show that more than 600 cases of violence were reported in 40 countries in the first 6 months of COVID-19 (Devi, 2020), and 58% of the respondents reported an increasing trend of MV (Thornton, 2022). The surging attacks in both developed and developing countries have brought physical or mental damage to HPs, which, as the final result, may cause declining productivity, the

shortage of labor supply among HPs, and the escalation of cost due to defensive medicine (Zhao et al., 2021).

Studies have separately documented the mental sequelae from MV and COVID-19 for HPs who experienced these events (Devi, 2020; Rossi et al., 2021; Tiesman et al., 2022). However, evidence quantifying the impact of their interaction, particularly the spillover effects on HPs exposed to different risks, is limited (Tiesman et al., 2022). First, exogenous risks against HPs may bring broader impacts except for those who experienced risk events (Zhao et al., 2021), which have been claimed to be heterogeneously distributed among HPs (Lai et al., 2020). Because of social culture since medical school, continuous education, and technical cooperation, HPs may hold intimate social relationships with counterparts (Cromwell et al., 2011). Information on MV and COVID-19 could spread quickly through formal and informal channels, where regional cases impose unexpected spillover effects (risk perception) among other HPs (Zhao et al., 2021). Second, the effects of the two and whether or how they may work together to generate a further and even greater negative impact on HPs, are relatively underestimated. Scholars have appealed to pay caution to the potential new challenge, but most studies present observational evidence for what happened within the COVID-19 context (Thornton, 2022; Tiesman et al., 2022). Whether and how the potential joint impacts can spill over remains unknown.

In mitigating COVID-19 and MV, the support of organizations (hospitals) has been academically and practically emphasized (Young et al., 2020; Haque, 2021; Liu et al., 2021a). In preventing MV and COVID-19, medical organizations have been documented as crucial actors (Yang et al., 2019). The absence of organizational support was associated with a higher likelihood of MV (Yang et al., 2019). As an unintended consequence, MV may trigger negative perceptions among HPs of their colleagues, patients, and hospitals (Lamothe and Guay, 2017), especially when the mitigation of COVID-19 is incorporated as a context (McKay et al., 2020; Liu et al., 2021a). Notably, most HPs work in public hospitals in China, which are special public facilities functioning as both government agencies and business organizations. Nevertheless, empirical evidence other than viewpoint for the crucial role of hospital support is rare (Young et al., 2020; Haque, 2021; Liu et al., 2021a).

This paper primarily examines the spillover effects of exogenous threats at the workplace on the MH of HPs. With data from a two-phase survey before and after the first outbreak of COVID-19 in one of the least developed provinces in China ($n = 10,901$), we explored the general impact of MV on HP's MH, how COVID-19 affects HP's MH, and whether it moderates the relationship between MV and MH. Secondly, we also examined the influence of organizational support on the relationship between MV/COVID-19 and HP's MH. HP's individual-level perception of support from their hospitals (e.g., effort-reward imbalance, job satisfaction, and organizational commitment) was used to denote organizational support when HPs face exogenous workplace risks.

## Method

### Setting

Our surveys were implemented in December 2019/2020 in Gansu (GS) Province, one of the least developed regions covering 14 prefecture cities in China. The population of Gansu is 25.02 million. Its capital Lanzhou (LZ), and nearby Tianshui (TS) are the top two cities, with 4.40 and 2.98 million of population. GS is a typical region with poor provision but high demand for healthcare. As Supplementary Figure A1 shows, the *per capita* GDP in GS was only approximately 50% of the national average; however, its *per capita* total health expenditure (THE) accounted for 80% of the national average. Although the provision of physical facilities such as beds (per 1,000 population) was improved, the shortage of physicians remained. Even so, MV increased if measured by the amount of annual lawsuit cases related to physician-patient conflicts.

We launched two waves of surveys on HPs (Supplementary Figure A2). The first-round survey (2019 survey) was implemented in December 2019 in TS. Questionnaires were collected onsite at hospitals. The sample size is 5,304, accounting for 28.7% of the HPs in TS (The total sample is 5,500, but 196 were excluded because of missing information). Approximately 29 mainstream hospitals were included in the survey. Participation in HPs was voluntary, anonymous, and untraceable. The second survey (2020 survey) was implemented in December 2020, 7 months after the first outbreak of COVID-19 (January–April 2020). Using snowball sampling, this survey collected information through WENJUANXING, a leading online survey organization in China. All 14 cities of Gansu were covered in this survey but with a focus on the capital city, LZ. Finally, 5,610 responses were collected, with a final sample of 5,597 after excluding those with missing information.

In the two-round surveys, we used the same questionnaire containing the four sections of social demographic information, self-reported disease condition, self-rated physical and mental health, and information on self-perceived organizational support. No information or hints about MV and COVID-19 were asked to mitigate self-selection bias and participant manipulation. Even though the two-round surveys constitute cross-sectional data, the occurrence of MV and the outbreak of COVID-19 are exogenous shocks to an individual HP, providing us with the conditions for quasi-natural experiments. In particular, the outbreak of COVID-19 was beyond expectations and inspired us to launch the second round of the survey. Then, we define the first-round study as Study I, in which we explored the impact of MV on the MH of HPs, and the second-round study as Study II, to estimate the potential effects of both MV and the COVID-19 pandemic on HPs' MH.

### Measurements

*MH.* We use self-rated mental health status to represent the MH of HP, given that it has been considered a vital tool in studies on MH and the official diagnostic manual of the American Psychiatric Association (APA, 2013). Self-rated physical and mental health is measured by the Self-Rated Health Measurement Scale (SRHMS). Xu et al. (2010) localized rich studies on self-rated health (WHO, 2008), and developed the Chinese version of the SRHMS. It is a 48-item, 10-point scale to measure individuals' physical, mental, and social health and has been broadly utilized to measure the self-rated health of HPs in China. We selected mental and social health items to denote the MH of HPs, mitigating potential concerns about cherry-picking outcome variables. Twenty-seven items were categorized as positive emotion (5 items), symptom and negative emotion (negative emotion for short, 7 items), cognitive function (cognition for short, 3 items), role activity and social adaptability (social adaptability for short, 4 items), social resource and social contact (social connection for short, 5 items), and social support (3 items) (Xu et al., 2010). Every item is rated on a 10-point and shares the same weight. Because social dysfunction is claimed to be closely associated with MH (Cornaglia et al., 2015), we include social health as a partial measure of MH. Finally, we averaged items

by the 6 dimensions with the same weight and obtained 6 MH indicators (details are provided in Supplementary Table A1). In this study, the Cronbach's alpha of scale is 0.939.

*MV.* Violent verbal or physical attacks on HPs are emerging globally and even worse in China. However, the measurement is challenging. Because of the long-term stigmatization from previous disgrace in the past two decades (Wang et al., 2020), many verbal or physical attacks on HPs were downplayed in China. Relevant data collection and impact, consequently, are limited and difficult. Scholars have adopted the intensity of attacks reported in news reports about specific events as a proxy alongside survey data (Zhao et al., 2021). It is a wise option but may vastly underestimate the incidence and impact of these events. Only the worst cases, such as the murder of HPs, can be publicly reported. Therefore, we use litigation records from the China Judgment Online System (CJOS) to denote the incidence of regional MV. CJOS is an official database of the Supreme People's Court of China, which digitally archives judiciary documents of criminal cases nationwide since 2013. The cases of MV with lawsuits imply that the intensity of conflicts is severe and not downplayed. Most of these cases in a relatively small region are easily known by most HPs; therefore, their impacts may be strong enough within-region. Even though this measurement inevitably underestimates the incidence frequency of MV events, it is a better choice than media reports and survey data when the impact of MV is estimated at the regional instead of individual level. The search criteria in the CJOS database follow similar studies (Cai et al., 2019). Finally, the number of cases in prefecture cities was collected as the proxy for the regional intensity of MV in 2019 and 2020. Panel D of Supplementary Figure A1 shows the variation trend (increasing) of lawsuit cases in GS in the past 5 years, and the regional distribution in 2019 and 2020 can be found in Supplementary Table A2. One assumption is that the number of cases is exogenous, and their impacts are homogeneous for individual HPs.

*COVID-19 exposure.* The construction of variables for COVID-19 spread is quite straightforward. Following the strategy of Liu et al. (2021b), we use the confirmed cases in 2020 for every prefecture city to indicate the regional spread intensity of COVID-19. The original data came from the health authority of GS. Supplementary Figure A2 shows that the first case of COVID-19 in GS was identified on January 23, 2020. Since January 25, 2020, GS has launched the highest-level response to the public health emergency, and this outbreak wave terminated in April 2020. In the later stage, only sporadic cases were confirmed until December 2020, when our first-round survey was launched, and 182 COVID-19 cases (including local and imported cases) were confirmed in GS. The regional distribution is reported in Supplementary Table A2.

*HP's self-perceived organizational support.* The measurement of organizational support is another challenge. Although organizational support security to prevent MV has increased since the Law of the People's Republic of China on the Promotion of Basic Medical and Health Care in 2020, heterogeneity is inevitable, and hospitals usually do not disclose their efforts. In this case, we use HP's objective perception to measure the extent of hospital support. We use three approaches to denote HP's self-perceived organizational support. The first is the *Effort-Reward-Imbalance model* (ERI), a classic proxy for occupational stress (Siegrist et al., 2004), including stress and burnout of HPs. We use the weighted ERI (Effort/Reward) to indicate the self-rated efforts and rewards for HPs in the organizations they belong. The second is job satisfaction using the three-dimensional scale (Hackman and Oldham, 1976), which captures workers' general satisfaction, intrinsic work

motivation, and special satisfaction. The last is the organizational commitment model of Allen and Meyer (1990). Following the authors' suggestion, we use the affective component of organizational commitment to measure HP's "emotional attachment to, identification with, and involvement in, the organization" (Allen and Meyer, 1990).

### Statistical analysis

The primary intention of this study was to estimate the spillover effects of MV and COVID-19 on the MH of HPs. The cross-sectional data in this study confine our implementation of strategies of other studies (Braghieri et al., 2022), but the features of MV and COVID-19 provide an ideal setting for natural experiments. For individual HPs, MV and COVID-19 could be exogenous, which is unexpected and not manipulated by others to a large extent. Otherwise, HPs cannot control the incidence of MV and COVID-19 under normal conditions if there is no intentional malpractice. Therefore, we argue that the empirical models shown in Supplementary Appendix A may be reliable, where three studies were designed to cross-validate the reliability (A detailed description of the analytic models is shown in Supplementary Appendix A).

In study I, we only estimate the impact of MV on MH of HPs using 2019 data. Due to the data limitation that no regional variation on MV can be observed for the 2019 data, we assume the homogenous risk exposure for individual HP and examine the variable consequences of the same risk brought to different HP groups. We divided HPs into three groups, physicians, nurses, and others, according to levels of risk exposure in the Chinese context, where the other HPs were treated as the reference group (Wang et al., 2020; Liu et al., 2023). Study II allows us to explore the regional variation in MV and COVID-19 exposure on HPs' MH with 2020 data. The potential impact of MV, COVID-19, and whether COVID-19 has exacerbated the impact of MV on HPs' MH are separately examined with the exploration of effects of MV, COVID-19, and their interaction on MH of different HP groups (physicians, nurses, and others) as Study I.

Although two of our surveys are cross-sectional, the before and after COVID-19 set allows us to compare its effects on HPs' MH using both 2019 and 2020 data as additional evidence. Following Azoulay et al. (2013), we use the coarsened exact matching (CEM) strategy to match the 2019 and 2020 data. CEM, a nonparametric matching approach, can conserve sample size with less dependence on regression models and is easier to reach the principle of consistency than regular match approaches such as propensity score matching (Iacus et al., 2009; Iacus et al., 2011). We mainly matched the 2019 data in TS with the 2020 data in LZ because they are the two largest cities in GS and have approximate levels of hospital distribution.

The same estimation strategies above (as shown in Supplementary Appendix A) were also used to investigate the effects of organizational support. First, the relationship between organizational support and MH of HPs was examined as a baseline reference. Second, the same three studies were employed to estimate whether organizational support can mitigate the adverse effects of MV, COVID-19, and their interaction on MH of HPs. The independent variables in equations (1)–(5) were replaced by their interaction terms with variables denoting organizational support. In all the estimations, Ordinal Least Regression is our primary approach.

## Results

### Descriptive statistics

Table 1 shows the descriptive statistics of data. Column (1) presents the results of the 2019 survey, while Column (2) shows summaries for the 2020 survey. Panel A averages individual-level variables, including gender, age, marriage status, education levels, professional titles, income levels, and the category of medical roles. In the 2019 survey, 81% of the respondents were female, and 28% were physicians. In the 2020 survey, though 80% of the respondents were female, the proportion of physicians increased to 39%. Panel B reports the averaged results of individual-level MH. The two waves of surveys show similar results for 6 indicators, where every indicator was weighted by item shown in Supplementary Table A1. Panel C of Table 1 shows the information on MV and COVID-19. In 2019, there were 22 lawsuit cases regarding MV, and no COVID cases were identified. In 2020, the 14 cities in GS averagely reported 17 lawsuit cases on MV and 13 confirmed COVID-19 cases. Panel D portrays the profile of indicators for organizational support. It reports a 1.1 (0.8, 1.4) score of ERI (1 denotes balance), implying a slight imbalance of attitude among HPs regarding their effort to and reward from the hospitals where they work. Job satisfaction is 66% (44%, 87%) and 67% (49%, 86%), and the corresponding organizational commitment levels are 62% (50%, 74%) and 70% (58%, 82%) for the 2019 and 2020 surveys.

### Effects of MV, COVID-19, and their interaction

*Study I.* Figure 1 (Supplementary Table A3, Panel A) illustrates the results estimated by Equation (1) in Supplementary Appendix A on MH. The first two rows present results of MV on HP's positive emotion, where a significant decrease of positive emotion can be observed, and the effect is more substantial for physicians (*coefficient:* $-0.22$ $(-0.26, -0.18)$, $p < 0.01$) than nurses (*coefficient:* $-0.07$ $(-0.11, -0.03)$, $p < 0.10$). The second two rows report the results of negative emotion. MV significantly increases HP's negative emotions, and the effects on physicians are also more significant. The rest rows in Figure 1 display the results of cognition, social adaptability, social contact, and social support. Most coefficients do not differ from zero, except for the adverse effects of MV on social adaptability (*coefficient:* $-0.09$ $(-0.12, -0.06)$, $p < 0.01$) and social support (*coefficient:* $-0.11$ $(-0.18, -0.05)$, $p < 0.10$). In general, the results of Study I suggest that exposure to MV may adversely deteriorate HP's MH, including emotional disorders and partial social health, and physicians are the most impacted group.

*Study II.* Estimates of $\beta$ in Equations (2), (3), and (4) in Supplementary Appendix A are visualized in Figure 2 (Supplementary Table A3, Panels B, C, and D). First, the regional variation of exposure to different levels of MV allows us to supplement the results in Study-I with more extensive evidence. The coefficients show the sizeable effects of MV on HP's MH. Except for no significant results of positive emotion, higher intensity exposure to MV increased HP's negative emotion (*coefficient:* 0.20 (0.18, 0.21), $p < 0.01$) and decreased their levels of cognition (*coefficient:* $-0.03$ $(-0.04, -0.02)$, $p < 0.05$) and social contact (*coefficient:* $-0.07$ $(-0.08, -0.06)$, $p < 0.01$). However, it is also found that this adverse shock increased HP's levels of social adaptability (*coefficient:* 0.08 (0.07, 0.08), $p < 0.01$) and social support (*coefficient:* 0.06 (0.05, 0.07), $p < 0.01$).

Second, high exposure to COVID-19 had the same trend but with fewer effects than MV. Specifically, the corresponding coefficients are *0.18 (0.16, 0.19)] (p < 0.01), −0.03 (−0.05, −0.02) (p < 0.05), 0.06 (0.05, 0.08) (p < 0.01), −0.06 (−0.07, −0.05)*

**Table 1.** Summary statistics of the 2019 and 2020 surveys

| | 2019 survey (n = 5,304) | 2020 survey (n = 5,594) |
|---|---|---|
| *Panel A: individual characteristics* | | |
| Gender | | |
| Male | 1,008 (19%) | 1,119 (20%) |
| Female | 4,296 (81%) | 4,475 (80%) |
| Age | 31.39 ± 7.83 | 33.94 ± 8.88 |
| Marriage | | |
| Married | 3,342 (63%) | 4,140 (74%) |
| Unmarried | 1856 (35%) | 1,399 (25%) |
| Divorced | 53 (1%) | 56 (1%) |
| Others | 53 (1%) | 0 |
| Education | | |
| Technical school | 318 (6%) | 224 (4%) |
| Diploma | 2,440 (46%) | 1,566 (28%) |
| Bachelor | 2,440 (46%) | 3,356 (60%) |
| Master or above | 106 (2%) | 448 (8%) |
| Professional title | | |
| Junior | 159 (3%) | 3,300 (59%) |
| Middle | 3,660 (69%) | 1,454 (26%) |
| Senior | 1,220 (23%) | 559 (10%) |
| Others | 265 (6%) | 280 (6%) |
| Income (RMB) | | |
| ≤3,000 | 1962 (37%) | 1,399 (25%) |
| (3,000, 6,000] | 2,917 (55%) | 3,021 (54%) |
| >6,000 | 424 (8%) | 1,175 (22%) |
| Role | | |
| Physicians | 1,485 (28%) | 2,182 (39%) |
| Nurses | 3,554 (67%) | 2,461 (44%) |
| Others | 265 (5%) | 951 (17%) |
| *Panel B: Mental health* | | |
| Positive emotion | 7·23 ± 1·95 | 7.32 ± 1.92 |
| Negative emotion | 5·28 ± 2·09 | 5·58 ± 2·03 |
| Cognition | 6·44 ± 1·84 | 6·47 ± 1·77 |
| Social adaptability | 7·51 ± 1·56 | 7·76 ± 1·53 |
| Social connection | 6·95 ± 1·80 | 6·79 ± 1·90 |
| Social support | 6·64 ± 1·93 | 6·59 ± 1·94 |
| *Panel C: MV, COVID-19, and cites* | | |
| Cases (MV) | 22 ± 0 | 17·00 ± 4·99 |
| COVID-19 | 0 | 13·00 ± 8·83 |
| Number of cities | 1 | 14 |
| *Panel D: Organizational support* | | |
| ERI | 1·1 ± 0·3 | 1·1 ± 0·3 |
| Satisfaction (Job) | 4·6 ± 1·5 | 4·7 ± 1·3 |
| Organization support | 3·1 ± 0·6 | 3·5 ± 0·6 |

*Note:* Categorical variables are expressed as % (n); continuous variables are expressed as the mean ± SD.

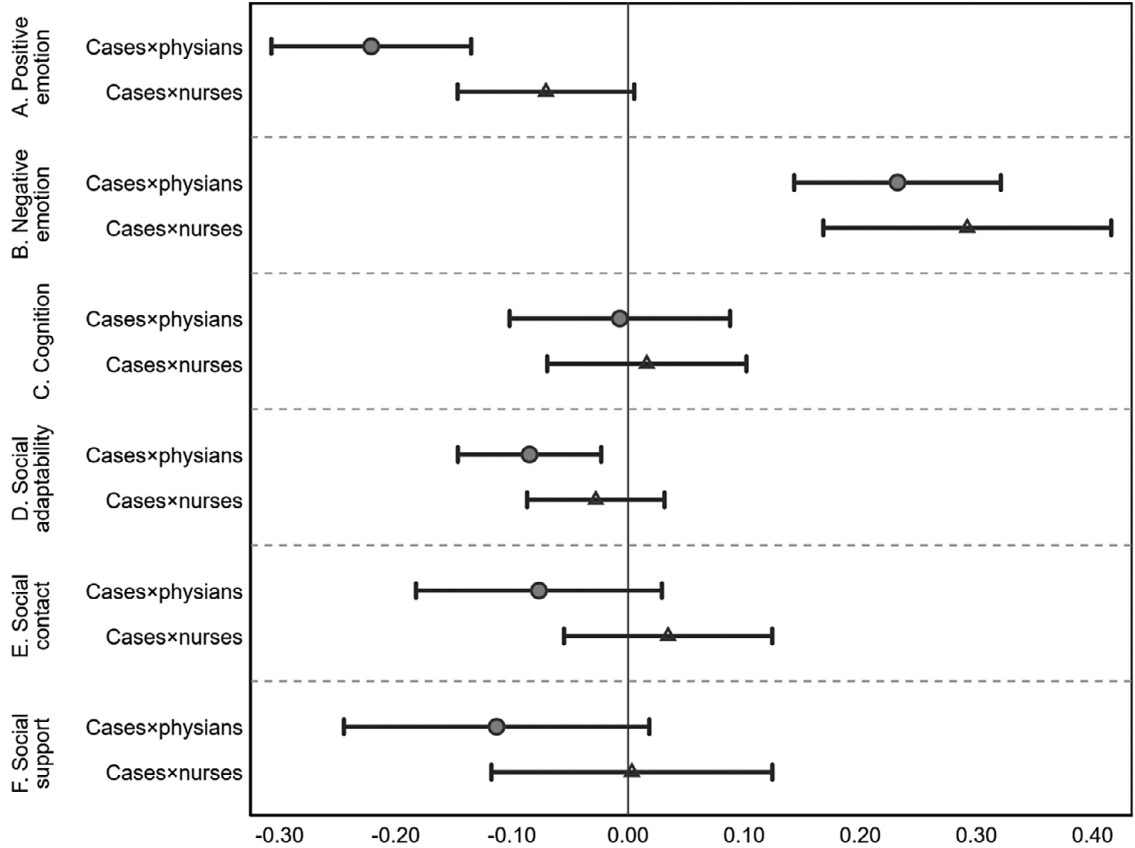

**Figure 1.** Impacts of MV on HP's mental health (Study I).
*Notes:* This figure report results from Equation (1) in Supplementary Appendix A; the spots are coefficients, and the solid lines indicate 95% confidence intervals. The baseline group is the HPs other than physicians and nurses. Cases = the lawsuit cases of MV.

($p < 0.01$), and 0.06 (0.05, 0.06) ($p < 0.01$) for negative emotion, cognition, social adaptability, social contact, and social support. This may imply that the outbreak of the COVID-19 pandemic brought tremendous psychological pressure on HP. An intuitive interpretation is that the high workload and risk exposure in the frontline of pandemic mitigation deteriorated HP's mental status and deprived them of regular contributions to family and social activities.

Third, the interaction effects of MV and COVID-19 present much more significant effects than either MV or COVID-19. Here the coefficients are 0.24 (0.22, 0.26) ($p < 0.01$), −0.04 (−0.06, −0.03) ($p < 0.05$), 0.08 (0.06, 0.10) ($p < 0.01$), −0.08 (−0.09, −0.07) ($p < 0.01$), and 0.07 (0.07, 0.08) ($p < 0.01$) for negative emotion, cognition, social adaptability, social contact, and social support, which are invariably greater than the coefficients of COVID-19 and MV separately. This suggests that the outbreak of COVID-19 may exacerbate the adverse effects of MV, given the documented impacts of MV.

The results of how MV and COVID-19 affect different HP groups are reported in Supplementary Figure A3 (and Supplementary Table A3). Study II allows the analysis of impacts from different exposure levels of MV and COVID-19 on physicians, nurses, and other HPs. The same increasing trend of adverse effects of MV, COVID-19, and their interaction can be observed for physicians and nurses, where the non-frontline HPs are also the reference group. Overall, physicians manifested the worst MH as

the results of Study I, and even more significant conditions can be observed in Study II.

*Study III.* Figure 3 (and Supplementary Table A4) shows the results from the matched data to isolate the impacts of COVID-19 additionally. The first row of Panels A to F reports the main results. The coefficients are −0.70 (−0.78, −0.63) ($p < 0.10$), 0.66 (0.64, 0.67) ($p < 0.05$), −0.84 (−0.85, −0.83) ($p < 0.01$), −0.39 (−0.40, −0.39) ($p < 0.01$), −1.12 (−1.13, −1.12) ($p < 0.01$), and −0.89 (−0.91, −0.88) ($p < 0.05$) for positive emotion, negative emotion, cognition, social adaptability, social contact, and social support, showing the same but much more significant effect sizes than those in Figure 2. These results further prove the significant but negative impacts of exposure to risk from COVID-19 on HP's MH.

The rest rows of Figure 3 replicated the interaction effect of COVID-19 and MV and the impacts of COVID-19 and its interaction with MV on different HP groups with matched data. Similarly, the trend of the same effect can be found. The outbreak of COVID-19 deteriorated the impacts of MV on frontline HP's MH, while physicians bear the most severe impacts. However, the magnitude of the effects is much greater. Notably, the insignificant results on positive emotion shown in Figure 2 become significant.

*Robustness checks.* All coefficients (Supplementary Table A5) report the same signs as our main results, although the statistical significance of positive emotion varied. Conclusively, our main results are robust to a set of checks.

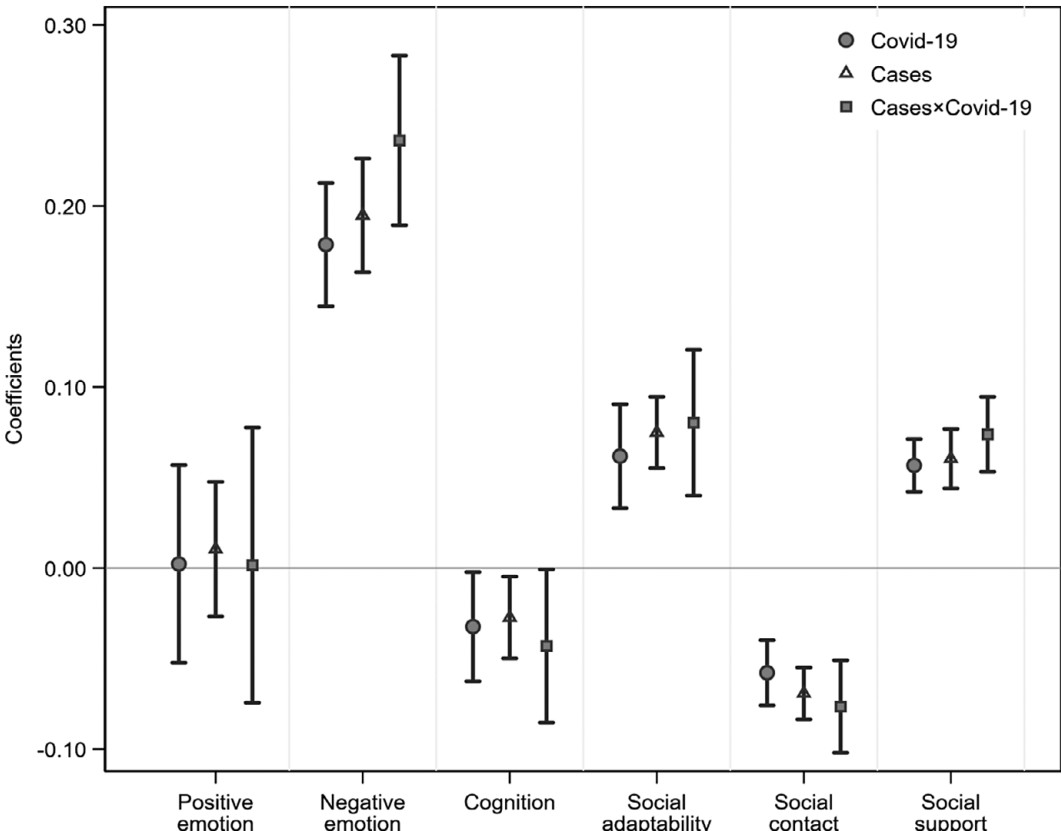

**Figure 2.** Impacts of MV and COVID-19 on HP's mental health (Study II).
*Notes:* This figure report results from Equations (2), (3), and (4) in Supplementary Appendix A, and the spots are coefficients, and the solid lines indicate 95% confidence intervals.
Cases = the lawsuit cases of MV, Covid-19 = the number of confirmed regional Covid-19 cases.

### *The role of organizational support*

Firstly, we directly explore the impact of organizational support on HP's MH using the same empirical strategies as the principal methodology while replacing the independent variables with proxies of organizational behaviors. The results are presented in Supplementary Table A6. From Panels A, B, and C, which show results for Studies I, II, and III, we can find that the increase in ERI significantly worsened HP's MH, while the increase in job satisfaction and organization commitment effectively improved HP's MH. These results suggest that the organization plays a crucial role in HP's MH, and the more support they receive from the hospitals they belong to, the better their MH will be. Of course, their MH deteriorates if they rate their devotion to the hospitals as much more than the reward they receive.

Second, we estimate the moderate effect of organizational behaviors on the established relationships between Covid-19 (MV) and MH. The results from Studies I and II are shown in Supplementary Tables A7 and A8. Results related to ERI, job satisfaction, and organizational commitment are presented in Panels A, B, and C. Supplementary Table A7 shows that the imbalance of ERI significantly moderates the adverse effects of MV on HP's MH, and physicians are the most affected group. However, the impacts of great job satisfaction and organizational commitment are significantly positive. Supplementary Table A8 reveals a similar effect, where the increase in ERI adversely affected the impact of COVID-19 and MV on HP's MH. At the same time, better job satisfaction and organizational commitment significantly improved the effect of COVID-19 and MV on HP's MH.

The results of Study III are shown in Supplementary Table A9. Panel A is the results of the treated COVID-19, while Panel B reports the interaction effects of organizational behaviors, COVID-19, and MV. Although most of the coefficients are statistically insignificant, the same trend of signs also can be observed. Conclusively, we may argue that the organizational behaviors of hospitals have a crucial impact on HP's MH. HP's bad or good perceptions of organizational support will significantly deteriorate or improve the effects of COVID-19 and MV on their MH.

### Discussion

Our results are consistent with previous literature but with certain specific contributions. First, we document the spillover effects of MV and COVID-19. Many studies found that COVID-19 or MV are adversely associated with mental health of HPs who experienced these events (Devi, 2020; Greenberg et al., 2020; Lai et al., 2020; Rossi et al., 2021; Ramzi et al., 2022; Tiesman et al., 2022). This study concretes these findings but shows that the adverse effects could spill to those aware of risk events but not victims. Zhao et al. (2021) found that the murder of physicians could generate cross-provincial effects on physicians in Chia. Our limitation of spillover effects into smaller regions is a substantial supplement.

Second, our results from regional measurement instead of self-reported screening measures of MV and/or COVID-19 complement previous studies on the adverse effects on HP's MH and provide new empirical evidence for their exacerbating interaction

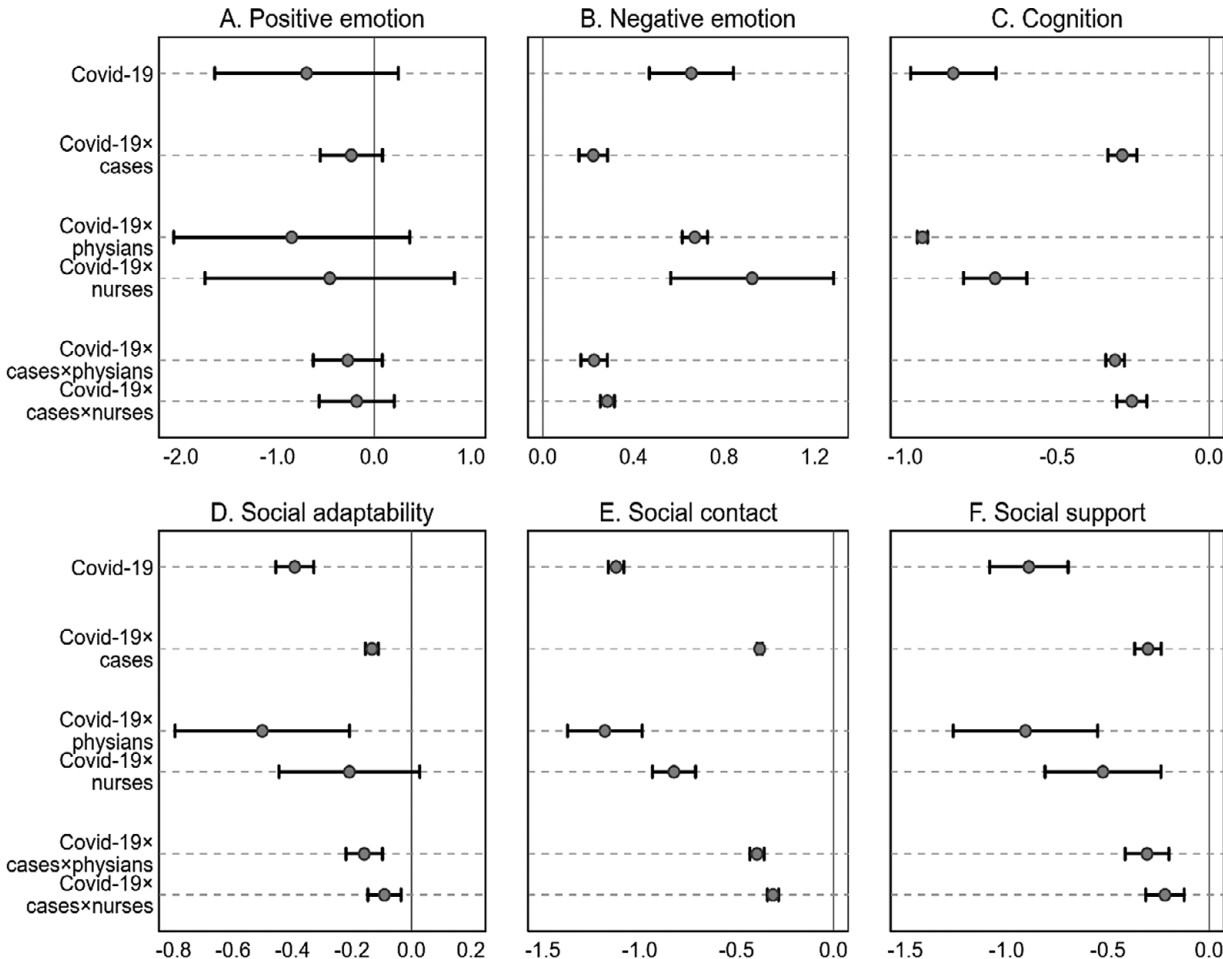

**Figure 3.** Impacts of COVID-19 on HP's mental health (Study III).
*Notes:* This figure reports results from Equation (5) Supplementary Appendix A, where the CEM approach matched the 2019 and 2020 survey data. The spots are coefficients, and the solid lines indicate 95% confidence intervals. The moderate effects of COVID-19 on the relationship between MV and HP's mental health were also estimated. Cases = the lawsuit cases of MV, Covid-19 = the number of confirmed regional Covid-19 cases.

effect on HP's MH. To our knowledge, this is the first study that explores the impacts of COVID-19 and/or MV on HP's MH using comparable data before and after the first outbreak of COVID-19 in China. We not only report the same adverse effects of MV and COVID-19 as previous studies (Lai et al., 2020; Rossi et al., 2021; Tiesman et al., 2022), but add empirical evidence that COVID-19 exacerbated the adverse effects of MV to similar comments or survey studies (Devi, 2020; Larkin, 2021; Ramzi et al., 2022; Thornton, 2022).

Third, we demonstrate the crucial role of organizational support of employees in the face of exogenous workplace threats. Teamwork has dominated modern healthcare production (Thibodeau et al., 2007), where HPs usually work in a specific organization, and organizational characteristics can create public value and influence HP's behavior (Thibodeau et al., 2007). However, most studies about the workplace threat of HPs, including risk from COVID-19 and MV, pay less attention to organizational support (Gillespie et al., 2010; Liu et al., 2021a). Although scholars have realized the importance of hospital organizational behaviors and have called for greater awareness, few empirical studies have examined their impacts (Walton et al., 2020; Young et al., 2020; Haque, 2021; Liu et al., 2021a). This study empirically examines the role of organizational (hospital) support and, most

importantly, obtains the expected results as previous academic appeals (Walton et al., 2020, Young et al., 2020, Haque, 2021, Liu et al., 2021a).

This study also has several policy implications. First, our proxies of MV and COVID-19 are bottom-line. As discussed in the research context, most MVs in China are not reported to the police and authorities, and only severe events have resorted to legal proceedings. So, speculation is that the adverse effects of MV and COVID-19 in the real world may be much more painful than the magnitude we estimated. Second, a reasonable but still unintended result is that physicians, but not nurses, bear the high risk. Previous studies in other countries contradict this (Gillespie et al., 2010). Referring to the related studies, it is easy to find the long-term stigmatization of physicians rather than nurses (Wang et al., 2020). We argue that HPs, particularly physicians in China, are vulnerable to institutional and historical changes and bear unnecessary stigmatization and hostility from patients and the public. The final highlight is the complicated organizational support for HPs in China. Because public hospitals owned by the government dominate the healthcare provision, where path-dependence from collectivism prevailed in the past decades, hospitals led by CCP provide houses, children's care, and even the resolution of daily trifles for HPs. Meanwhile, the public welfare

goal of public hospitals asks HPs to scarify themselves to pitch into mitigation when epidemics like COVID-19 outbreak. However, when MV happens, hospitals may betray their employees because they fear the emergence of public events. Since public hospitals have been criticized for profit-seeking behaviors for years, hospitals and authorities usually take the patient's side when conflict happens between HPs and patients because of the need to maintain social stability.

Despite the new evidence, this study faces inevitable limitations. In the second survey, we cannot track the same HPs in the first round survey. Although the match strategy is used, the explaining power of non-panel data may be affected. Also, we only report the situation of one province, one of the least developed regions in China, with poor healthcare provision. It can represent certain information in China but still face generalizability issues. Finally, as we have mentioned, MV and COVID-19 are measured by lawsuit cases and confirmed cases, which may underestimate the actual adverse impacts. However, we have argued to report a bottom-line situation. However, nothing about the impact of MV and COVID-19 was asked in the survey, which largely excluded the potential of bias from self-selection. At least, the bottom-line situation has already reported astounding adverse effects. Could the real-world scenario be worse? This question awaits to be answered by future studies.

## Conclusion

The unique finding is that MV and COVID-19 are also related to the social behaviors of the HP, and the outbreak of COVID-19 adversely moderates, also deteriorates the spillover effects of MV on HP's MH. Besides, our results suggest that physicians, but not nurses reported in other studies, are the most affected group, and the frontline workers bear the greatest risks. Finally, we find that HP's MH from COVID-19, MV, and their interaction are all worse when they perceive less organizational support. On the contrary, greater perceived support from hospitals can significantly mitigate the adverse effects of COVID-19, MV, and their interaction on MH for HPs.

**Open peer review.** To view the open peer review materials for this article, please visit http://doi.org/10.1017/gmh.2023.65.

**Supplementary material.** The supplementary material for this article can be found at http://doi.org/10.1017/gmh.2023.65.

**Data availability statement.** The datasets used and/or analyzed during the current study are available from the corresponding author on reasonable request.

**Author contribution.** Ning Liu: Main author, responsible for data analysis, writing, and revising. Hong Qian: Supported for the data analysis, data collection, and revising. Benzhong Zhang & Jing Guo: Corresponding author, responsible for study design, revising, and submitting.

**Financial support.** This work was supported by Humanities and Social Science Foundationof Ministry of Education in China [21YJC630082], Key Researchand Development Plan of Gansu Province, China [20YF8GA068], andthe Fundamental Research Funds for the Central Universities, China [2022jbkyjd002].

**Competing interest.** The authors declare no conflict of interest.

**Ethics standard.** The study protocol was approved by the Ethics Committee, the First Hospital of Lanzhou University (LDYYLL2023–61). All participants gave informed consent after being informed about the aims of the survey and joined the study voluntarily. All methods were carried out in accordance with relevant guidelines and regulations.

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
