## [Reviewer Report]

Dear Editor,

The increasing violent attacks against health professionals (HPs) at workplaces (or medical violence, hereinafter MV) like hospitals have gained global concern. Since the outbreak of COVID-19, the risk of violence in hospitals on HPs has seriously increased. The surging attacks in both developed and developing countries have brought physical or mental damage to HPs, which, as the final result, may cause declining productivity, the shortage of labor supply among HPs, and the escalation of cost due to defensive medicine.

With data from a two-phase survey before and after the first outbreak of COVID-19 in one of the least developed provinces in China (n=10901), we explored the general impact of MV (medical violence) on HP’s MH (mental health), how COVID-19 affects HP’s MH, and whether it moderates the relationship between MV and MH. Secondly, we also examined the influence of organizational support on the relationship between MV/COVID-19 and HP’s MH. HP’s individual-level perception of support from their hospitals (e.g., effort-reward imbalance, job satisfaction, and organizational commitment) was used to denote organizational support when HPs face exogenous workplace risks. 

We submitted the manuscript to the Global Mental health for the following reasons:

To the best of our knowledge, this study is the first use of micro-macro data systematically estimating the spillover effects of violent attacks, COVID-19 cases, and their interaction, on the mental health of health professionals in China. It also documents the importance of hospital support in mitigating mental health deterioration in workplaces. Using data from one of the least developed provinces in China, we provide empirical evidence of adverse spillover effects of violent attacks, COVID-19, and their interaction on the mental health of health professionals in China, where support from hospitals is crucial. Most importantly, we estimate a bottom-line situation in China, and then a worse national situation can be speculated.

Our findings support studies discussing the adverse effects of violent attacks or COVID-19 on the mental health of health professionals. Particularly, we empirically evidence that COVID-19 can deteriorate the adverse effects of violent attacks, where the adverse effects can spill but are not limited to those who bear the damage. In mitigating and recovering from the adverse spillover effects on mental health of health professionals incurred by violent attacks or COVID-19 and their interaction, support from hospitals is another crucial channel other than individual-level interventions. Our experience from one of the least developed provinces of China may be nationally or even globally generalized.

All authors have contributed significantly to the paper and approved this submission. There are no conflicts of interest in this manuscript. This original research has not been submitted elsewhere.

Yours sincerely,

Jing Guo, Department of Health Policy and Management, School of Public Health, Peking University, 38 Xueyuan Road, Beijing, 100191, P. R. China. Email: jing624218@163.com

---

## [Reviewer Report]

This study aimed to examine the spillover effects of violent attacks, COVID-19 exposure, and their interactions on health professionals’ mental health, and the role of organizational support in their relationships in China. The survey results showed Violent attacks and COVID-19 are negatively associated with the mental health of health professionals, and the outbreak of COVID-19 adversely deteriorates the spillover effects of violent attacks.

The whole research idea is clear, the technical route is reasonable, the methods used are appropriate, the data processing is reliable, and the conclusions are credible. The writing is standardized, and the text is fluent, which meets the requirements of academic paper writing.

---

## [Reviewer Report]

The aims of this study are to examine the spillover effects of violent attacks, COVID-19 exposure, and their interactions on health professionals’ mental health, and the role of organizational support in their relationships in China:

What type of violence, who perpetrates it, in what period of

study, which supports the organization.

What does it mean “the spillover effects of exogenous”threats at the workplace on the

MH of HPs.

The study population, space and time are not clear.

A two-phase survey data (n=10,901) before and after the first outbreak of COVID-19. In this aspect, must understand that the first outbreak was accompanied by uncertainties, for the health professionals and the population. Regarding the support organizations, they were exposed to same topic. The question is, could this overestimate the results?

It is important to mention whether the assumptions such as no multicollinearity, no Heteroscedasticity, no autocorrelation and normality of waste or errors. This is to prevent the results from being considered spurious.

---

## [Reviewer Report]

The manuscript is sufficiently clear regarding the context, background, methods, results and conclusions. There are minor observations from one of the reviewers, but the change would not be a fundamental contribution. Therefore, it is considered suitable for publication.